# Antimicrobial Quantitative Relationship and Mechanism of Plant Flavonoids to Gram-Positive Bacteria

**DOI:** 10.3390/ph15101190

**Published:** 2022-09-27

**Authors:** Ganjun Yuan, Xuexue Xia, Yingying Guan, Houqin Yi, Shan Lai, Yifei Sun, Seng Cao

**Affiliations:** 1Biotechnological Engineering Center for Pharmaceutical Research and Development, Jiangxi Agricultural University, Nanchang 330045, China; 2Laboratory of Natural Medicine and Microbiological Drug, College of Bioscience and Bioengineering, Jiangxi Agricultural University, Nanchang 330045, China

**Keywords:** flavonoid, lipophilicity, MIC, relationship, bacteria, cell membrane

## Abstract

Antimicrobial resistance (AMR) poses a serious threat to human health, and new antimicrobial agents are desperately needed. Plant flavonoids are increasingly being paid attention to for their antibacterial activities, for the enhancing of the antibacterial activity of antimicrobials, and for the reversing of AMR. To obtain more scientific and reliable equations, another two regression equations, between the minimum inhibitory concentration (MIC) (*y*) and the lipophilicity parameter ACD/LogP or LogD_7.40_ (*x*), were established once again, based on the reported data. Using statistical methods, the best one of the four regression equations, including the two previously reported, with regard to the antimicrobial quantitative relationship of plant flavonoids to Gram-positive bacteria, is *y =* −0.1285 *x*^6^ + 0.7944 *x*^5^ + 51.785 *x*^4^ − 947.64 *x*^3^ + 6638.7 *x*^2^ − 21,273 *x* + 26,087; here, *x* is the LogP value. From this equation, the MICs of most plant flavonoids to Gram-positive bacteria can be calculated, and the minimum MIC was predicted as approximately 0.9644 μM and was probably from 0.24 to 0.96 μM. This more reliable equation further proved that the lipophilicity is a key factor of plant flavonoids against Gram-positive bacteria; this was further confirmed by the more intuitive evidence subsequently provided. Based on the antibacterial mechanism proposed in our previous work, these also confirmed the antibacterial mechanism: the cell membrane is the major site of plant flavonoids acting on the Gram-positive bacteria, and this involves the damage of the phospholipid bilayers. The above will greatly accelerate the discovery and application of plant flavonoids with remarkable antibacterial activity and the thorough research on their antimicrobial mechanism.

## 1. Introduction

Antimicrobial resistance (AMR) has become a serious threat to the public health; meanwhile, the COVID-19 pandemic has further accelerated this global problem [1]. So, new antimicrobial agents are desperately needed [2,3]. After antibiotics have been used for the treatment of bacterial infection, most of them will also bring about some adverse reactions and eventually be resistant in the clinic [4]. However, some plant metabolites with moderate antimicrobial activities [5], being nontoxic to the human body, can enhance the antibacterial activity of some antibiotics, and even reverse the AMR [6,7]. Among them, plant flavonoids have received close attention [8,9,10,11,12]. Some of their structure–activity relationships against bacteria were summarized in various degrees [7,8,13,14], together with some sporadic ones [15,16]. In addition, the quantitative structure–activity relationship (QSAR) analyses for 30 prenylated (iso)flavonoids against *Listeria monocytogenes* and *Escherichia coli* were performed, respectively, with an accuracy of 71–88% [17]. However, a universal and systematic conclusion remains unclear due to the extensive structural diversity of plant flavonoids, and some of the conclusions are even contradictory [7,8,13,14].

In our previous work [18], two regression equations were established for calculating the antibacterial activities of plant flavonoids towards Gram-positive bacteria, based on the data pairs, consisting of the physicochemical parameter ACD/LogP or LogD_7.40_ and the minimum inhibitory concentration (MIC, an indicator of antibacterial activity), of 66 reported flavonoids [19,20,21,22,23,24]. Subsequently, these two equations were further verified by the data pairs of another 68 reported flavonoids [6,25,26,27,28,29,30] and presented the accuracy of 85.3%. Combined with the literature analyses, it concluded that the lipophilicity is a key factor for flavonoids against Gram-positive bacteria and that the cell membrane is the major action site [18].

To obtain more scientific and reliable regression equations for the prediction of the MIC values of plant flavonoids, those data, as a greater sample, were reanalyzed, and two regression equations were reestablished. Using statistical methods, a regression equation with a larger correlation coefficient (*r*) of 0.9703 eventually proved to be the best one for fitting the correlation between the antibacterial activity (MIC) and the lipophilicity (LogP). This equation has shown to be more accurate and more reliable and can be practically considered as the quantitative relationship of plant flavonoids against Gram-positive bacteria. Moreover, the regression curves between the log_10_ (MIC) (*y*) and the LogP (or LogD_7.40_) value (*x*) provide more intuitive evidence for the correlations between the antibacterial activity and the lipophilicity and for the antibacterial mechanism of the plant flavonoids acting on the cell membrane. The above are diagrammatically presented in Figure 1.

## 2. Results

### 2.1. Structure, Antibacterial Activity, and Physicochemical Parameters

The one hundred and thirty-four flavonoids published in the previous work [18], from twelve papers [19,20,21,22,23,24,25,26,27,28,29,30], were reorganized, and 92 compounds were screened out, according to the procedure in the methods section, for subsequent regression analyses. These flavonoids involve eleven subclasses, which mainly include flavones, dihydroflavones, flavonols, dihydroflavonols, isoflavones, dihydroisoflavones, dihydroisoflavane, and chalcones. The serial numbers of these compounds remain unchanged and correspond to those in the previous work [18]. Their physicochemical parameters (ACD/LogP and LogD_7.40_) and antimicrobial activities (MICs) are listed in Table 1. If possible, the average MIC or MIC_90_ of a certain flavonoid to different pathogenic bacteria was considered as its MIC. In other cases, the MIC of a certain flavonoid to pathogenic bacteria was processed according to the rules in the methods section.

### 2.2. Regression Equation between the MICs and the Physicochemical Parameters

The regression analyses for the MICs (*y*) to Gram-positive bacteria and the physicochemical parameters LogP or LogD_7.40_ (*x*) of these flavonoids were achieved. Two regression curves are shown on Figure 2; their regression equations were established and are shown in Figure 2 and in Table 2, together with their correlation coefficients (*r*).

From Figure 2, the characteristics of these two regression curves were similar to those established from the 66 flavonoids [18]. However, they presented larger *r* values (Table 2) and thereby indicated more significant correlations between the physicochemical parameter LogP or LogD_7.40_ and the MIC of the plant flavonoids to Gram-positive bacteria, especially for that between the parameter LogP and the MIC, which presented the largest *r* value of 0.9703 (Table 2). Thereby, these two equations have a greater potency in proving that the antibacterial activities of plant flavonoids to Gram-positive bacteria are close related to their lipophilicities.

### 2.3. Antimicrobial Quantitative Relationship

Including the two regression equations reported [18], four regression equations were established for fitting the correlation between the antimicrobial activity (MIC) and the physicochemical parameter (LogP or Log D_7.40_). To compare the goodness of fit, two statistical parameters, the coefficient of determination (*R^2^*), and the residual standard deviation (*s*) were calculated, respectively, for these four equations and presented in Table 3. Generally, the closer the *R****^2^*** is to 1, the higher the goodness of fit and the closer the calculated value is, on the whole, to the actual one. The smaller **s** is, the smaller the mean deviation between the calculated value and the actual one. From Table 3, the largest value of the *R^2^* (0.9413) and the smallest value of *s* (68.1127) indicated that Equation (1) (Table 2) is the most reliable and the best one for fitting the quantitative relationship between the LogP and the MIC of the plant flavonoids to Gram-positive bacteria. Considering that the accuracy predicted from Equation (4) is approximately 85.3% [18], the above sufficiently indicated that that which is from Equation (1) has greater accuracy and is more than 85.3%, as it has a larger ***R^2^*** value and a far lower **s** value than Equation (4). Therefrom, the MICs (*y*) of most plant flavonoids to Gram-positive bacteria can be more accurately calculated from this equation by substituting their LogP values (*x*) (calculated by ACD/Labs 6.0).

### 2.4. Regression Equation between the Log_10_(MIC) and the Physicochemical Parameter

Based on the regression equations previously established and the literature analyses, the antibacterial mechanism of the plant flavonoids acting on the cell membranes of Gram-positive bacteria was proposed [18]. To more intuitively observe the correlation between the antibacterial activity and the lipophilic parameters, the regression analyses for the common logarithm (log_10_) of the MIC (*y*) to Gram-positive bacteria and the LogP or LogD_7.40_ (*x*) of these plant flavonoids were further performed. Their regression curves and regression equations, with the *r* values of 0.8040 and 0.8212, are shown, respectively, in Figure 3.

Both *r* values are greater than 0.27, which is the critical value when α is set at 0.01 and the sample number is ninety-two. This indicates that there is a very significant correlation between the log_10_(MIC) and the LogP or LogD_7.40_. Along with the increase of the LogP or LogD_7.40_ value in an approximate range from 2.0 to 8.0, the log_10_(MIC) value decreases, i.e., the antibacterial activity increases. These, more intuitively, demonstrated that the antibacterial activities of plant flavonoids to the Gram-positive bacteria are directly related to their lipophilicities.

## 3. Discussion

Flavonoids are an important class of secondary metabolites widely distributed in various parts of the plant, and so far, approximately 10,000 compounds have been discovered. These compounds have various bioactivities, such as antioxidation, antibiosis, an estrogen-like effect, and the prevention and treatment of cardiovascular diseases [6,31,32]. After some of them were discovered to have the potency to enhance the antibacterial effect of some antibiotics and/or even reverse the AMR [6,7], their antibacterial activities have been increasingly receiving close attention [8,9,10,11,12]. However, the antimicrobial activities of most flavonoids remain unknown after being discovered. Here, two equations between the lipophilicity (LogP or LogD_7.40_) and the antimicrobial activity (MIC) were established and verified by *r*-test according to the statistical analyses. Comparing the goodness-of-fit of four equations, including the two reported ones [18], Equation (1) is the best one for calculating the MICs of plant flavonoids to Gram-positive bacteria, and the predicted accuracy is at least 85.3%. This equation can be widely used for many works related to the antimicrobial research of plant flavonoids: (1) the antimicrobial MICs of a larger amount of plant flavonoids already reported can be calculated and predicted, and therefrom, the plant flavonoids with the remarkable antimicrobial activity could be quickly screened from the flavonoids databases if one wanted; (2) based on the correlations between the lipophilicity and the antimicrobial activity and between the chemical structure and the lipophilicity, the antimicrobial activity can be quickly narrowed into a range after a compound is identified; (3) furthermore, the MIC values of the plant flavonoids to Gram-positive bacteria can be quickly predicted after they are isolated and identified, which would help to quickly target the desired one and simplify the MIC test; (4) as a good reference and guide, it will help to modify and optimize the chemical structure of plant flavonoids for potent antimicrobial agents; and (5) it can also provide a good reference for the structural modification and optimization of the plant flavonoids and reduce trial and error. All these will save a large amount of workload and human and material resources for the discovery of potent antimicrobial agents.

Based on the previous report [18], here the correlation between the ACD/LogP or LogD_7.40_ values and the MICs to Gram-positive bacteria of the plant flavonoids was further proved by a larger sample (*n* = 92), and both *r* values of the two regression equations were, respectively, 0.9703 and 0.9462, larger than those previously published. Thereby, these more powerfully proved that there is a direct correlation between the lipophilicity and the antibacterial activity of plant flavonoids. The statistical analyses, including the calculation and comparison of ***R****^2^* and ***s*** (Table 3), concluded that equation (1), as *y* = −0.1285 *x*^6^ + 0.7944 *x*^5^ + 51.785 *x*^4^ − 947.64 *x*^3^ + 6638.7 *x*^2^ − 21,273 *x* + 26,087, is the most scientific and reliable and is the best one. Specifically, the predicted value is more accurate and closer to the actual one, and/or the acceptable probability of the predicted MIC value is higher, at more than 85.3%, according to the same rule stating that the predicted MICs ranging from 1/4× to 4× the determined ones were acceptable [18]. More importantly, this equation was established from the data of eleven flavonoid subclasses, including seven main ones, while the equations previously reported were from those of three flavonoid subclasses. Thereby, the above together indicated that Equation (1) is more widely applicable and can be considered as the antimicrobial quantitative relationship of plant flavonoids to Gram-positive bacteria. Simultaneously, it can present an accuracy of approximate 94% (Table 3) according to the statistic principle, which is higher than the accuracy of 71–88% predicted from the QSAR of prenylated (iso)flavonoids against test bacterial isolates [17]. Moreover, Equation (1) can be at least used for the MIC calculation of eleven flavonoid subclasses against most Gram-positive bacteria, while the QSAR can only be used for that of prenylated (iso)flavonoids against two bacterial isolates.

In addition, here the correlations between the ACD/LogP or LogD_7.40_ values and the MICs to Gram-positive bacteria of plant flavonoids were further proved by a lager sample (*n* = 92), and both *r* values (0.9703 and 0.9462) of the two regression equations were larger than those previously published. Thereby, both of the two equations were better and more scientific in proving that there is a direct correlation between the lipophilicity and the antibacterial activity of plant flavonoids. The statistical evaluation procedure, including the calculation and comparison of *R^2^* and ***s*** (Table 3), concluded that Equation (1), as *y* = −0.1285 *x*^6^ + 0.7944 *x*^5^ + 51.785 *x*^4^ − 947.64 *x*^3^ + 6638.7 *x*^2^ − 21,273 *x* + 26,087, between the LogP value (*x*) and the MICs (*y*), is the most reliable and the best one. More importantly, this equation was established on the data of eleven subclasses of flavonoids, including seven main subclasses, while those previously published were generated from the data of three subclasses of flavonoids. Thereby, the above together indicated that Equation (1) is more scientific, reliable, and universal and can be considered as the antimicrobial quantitative relationship of plant flavonoids to Gram-positive bacteria.

As many factors involving the methods and details of the MIC test may have an influence on the experimental MIC value, the antibacterial activities of a compound to different pathogens are usually varied [18]. Therefore, the tested MICs would fluctuate within a reasonable range, especially from 1/2× to 2× the actual values [18], since the MICs were generally tested by the double dilution method. Simultaneously, the LogP value calculated by soft ACD/Labs 6.0 generally presents as a range. Thereby, the determined MICs would probably range from 1/2× to 2× the predicted one or more probably from 1/4× to 4× the predicted one. Based on these, the MIC (more accurately, MIC_90_) for a certain compound of flavonoids to Gram-positive bacteria can be calculated by substituting its ACD/LogP value (*x*) into Equation (1). Furthermore, the minimum MIC of plant flavonoids to Gram-positive bacteria can be predicted as approximately 0.9644 μM, and at this time, the LogP value is equal to about 7.055. Considering that the experimental MICs would fluctuate, the minimum MIC tested would more probably range from 0.24 to 0.96 μM.

The MICs of most plant flavonoids to Gram-positive bacteria can be correctly calculated from this equation, even if those flavonoid subclasses were not included when Equation (1) was established. For example, the MIC of α-mangostin, a xanthone from mangosteen, against Gram-positive bacteria was calculated as 8.16 μM (3.35 μg/mL), and so, it is deduced that the MICs tested would fall into the range from 0.84 to 13.4 μg/mL. This is, by and large, consistent with the determined MIC value of 1 or 0.5 μg/mL (0.5 μg/mL for MIC_90_) [33] and is also approved by the antibacterial tests repeated by two students at different times on S. aureus ATCC 25923 in our laboratory (0.5, 1 or 2 μg/mL for the MICs). Of course, a few of the plant flavonoids, such as baicalein, a rare 5,6,7-trihydroxyl flavonoid from *Scutellaria baicalensis*, probably present incorrect predictions [34]. However, the structural modification of baicalein for increasing the lipophilicity of molecules will increase the antibacterial activity [35]. This indicated that the correlation between the antibacterial activity and the lipophilicity is also suitable for baicalein, an ortho-trihydroxyl flavonoid.

As there were few data pairs in the LogP value range from 7.4 to 8.9 (Figure 2), the reliability of the calculation is possibly lower at this moment. It was already confirmed that the lipophilicity is a key factor of plant flavonoids against Gram-positive bacteria [18], while the influence of the dissociative state of plant flavonoids on their lipophilicities would gradually increase, along with the increase in the lipophilicity of plant flavonoids. Thereby, the LogD_7.40_, as the LogP at pH 7.40, is better to reflect the actual state of a compound in the medium of MIC determination, especially when the LogP value is large enough. Considering this, the parameter LogD_7.40_ should be more scientific and reliable than the LogP for fitting the correlation between the lipophilicity and the MIC, when the LogP value is more than 7.4. This was also supported by the change tendency of the two regression curves and the data pairs of the LogP values from 7.4 to 8.9 (Figure 2), as it is less possible for the regression curve to appear to drop twice according to the antimicrobial mechanism of plant flavonoids, especially when there is already a concave curve with a similar goodness of fit between the LogD_7.40_ and the MIC. Thereby, it should be more accurate to calculate the MIC values from the logD_7.40_ values using Equation (2), when the LogP values range from 7.4 to 8.9. It is probable that when the LogP values of the plant flavonoids are more than 8.9, this equation can be still used for the crude calculation for their MIC values against the Gram-positive pathogenic bacteria.

Similar to previous analyses [18], according to a similar procedure, the more reliable regression equations for a certain subclass of flavonoids, with a larger *r* value, can also established for the more accurate calculations for the MICs and for the structural modification and optimization of the plant flavonoids. It is worth noting that this equation is not necessarily suitable for the antibacterial calculation of all structural derivatives from plant flavonoids, especially those introducing heteroatoms, such as nitrogen and halogen.

In addition, there are some differences among the LogP or LogD_7.40_ values calculated by various software. As the LogP or LogD_7.40_ values in the equations were calculated by software ACD/Labs 6.0, both the lipophilic parameters must be calculated by the software ACD/Labs 6.0 (or updated edition) when the equations are applied. As previously reported [18], the correlations between the MIC and the LogP were not so significant and reliable if the LogP was calculated by ChemBioDraw Ultra 12.0 (CambridgeSoft Corporation, USA). Although the calculations of the lipophilic parameters are relatively mature, it is worth noting that some factors were still not considered, such as the stereochemistries of the chiral centers. Fortunately, plant flavonoids have few chiral centers, except for their pyran or furan rings, if they have even them, on which the chiral centers generally present identical stereochemistries. Thereby, the influence from some factors not considered will be reduced.

To better apply the antimicrobial quantitative relationship in practice, the main relationships between the lipophilicity (LogP) and the structure of the plant flavonoids, together with their consistency with some of the structure–activity relationships of the reported plant flavonoids [7,8,13,14,15,16,17], are presented in Table 4, in which some novel structure–activity relationships are also proposed. As everyone knows, the lipophilicity is influenced by many factors, such as the molecular structure and the pH environment, and the former also includes various substituent groups and their positions, etc. Some main factors contributed to the lipophilicity of the plant flavonoids, including the structural skeleton ring C, the hydroxyl groups, and the isopentenyl chains, and are presented in Table 4. Among them, the introduction of isopentenyl groups into rings A or B is the most important one, and it can remarkably enhance the LogP value and lead to the increase in the antimicrobial activity. Usually, this would mask the influences from other factors. However, the LogP value would sharply reduce when the hydroxylation occurred for the double bond of the isopentenyl groups, and thereby, it is deduced that the antimicrobial activity would remarkably reduce. These are completely consistent with the experimental MICs reported [8,18] and can be considered as novel structure–activity relationships of plant flavonoids against Gram-positive bacteria (Table 4). It is likely that the above are responsible for the confused, unsystematic, and even contradictory structure–activity relationships (SARs) of the plant flavonoids [7,8,13,14,15], especially for the effect of hydroxyls at the structural skeletons and their methylations on the antibacterial activities of the plant flavonoids. Moreover, the numbers and subclasses of plant flavonoids used for the establishments of most reported SARs are different and limited, which might be another reason for the confused and even contradictory SARs. Conversely, this further confirmed that the correlation between the lipophilicity and the antimicrobial activity of plant flavonoids, being established from 92 flavonoids including eleven subclasses, is scientific and reliable.

It is worth noting that the lipophilicity reflects the comprehensive characteristics of the whole molecular structure, while the traditional structure–activity relationships usually describe the contribution of a certain group and its position in the molecular structure to the antibacterial activity. As plant flavonoids include many structural subclasses, it is very difficult to conclude the universal structure–activity relationships, and different subclasses generally present different structure–activity relationships. Thereby, the simple summary of the structure–activity relationship of plant flavonoids against pathogenic bacteria easily leads to confused or inappropriate conclusions and even some contradictory results. This was also confirmed by the structure–activity relationships summarized from different laboratories [7,8,13,14]. As concluded above, the lipophilicity is the key factor responsible for the antimicrobial activity of plant flavonoids, including various subclasses, and therefrom, a universal quantitative relationship can be established. Thereby, these differences from different laboratories would be likely eliminated, especially that stating that the predicted MICs ranging from 1/4× to 4× the determined ones were acceptable [18]. Furthermore, the more accurate MIC values can be predicted from the calculated ones, with the help of some of the structure–activity relationships reported in [7,8,13,14] and proposed in Table 4.

As a previous work suggested [18], the cell-membrane is the major site of plant flavonoids acting on Gram-positive bacteria and likely involves the disruption or damage of the phospholipid bilayers. This was further supported by recent reports [15,17,33]. Here, more reliable regression equations further proved that the lipophilicity is a key factor responsible for the antibacterial activity of plant flavonoids to Gram-positive bacteria. Combined with previous work [18], the regression analyses for the correlation between the log_10_(MIC) and the LogP or LogD_7.40_ more intuitively confirmed the antibacterial mechanism of plant flavonoids acting on the cell membrane of Gram-positive bacteria. As previously pointed out [18], many other antibacterial mechanisms were mentioned in recent reviews [6,8,10], while most experiments were performed for the influence on the in vitro determination of enzyme activities [36,37], the molecular docking of plant flavonoids with various synthases [36,38], and the proteomic change without the intracellular verification and the consideration of whether the chicken or the egg came first [39]. In addition, the authors concluded that some mechanisms other than DNA gyrase inhibition may play a role in the antibacterial activity of flavonoids [29]. Therefore, together with many previous works [15,17,18,33], it is undoubted that the cell membrane is the main region of plant flavonoids acting on Gram-positive bacteria and likely involves the disruption or damage of phospholipid bilayers or some others. Therefrom, the prior direction for clarifying the mechanism of flavonoids against Gram-positive bacteria is ascertained.

Recently, many antimicrobial mechanisms and SARs of plant flavonoids against Gram-negative bacteria have been reported [17,40,41]. As this research focused on the antimicrobial quantitative relationships and mechanisms of the plant flavonoids against Gram-positive bacteria, those of the plant flavonoids against Gram-negative bacteria were not discussed here. In addition, it can be deduced that there are likely to be different antimicrobial mechanisms for plant flavonoids against Gram-positive and Gram-negative bacteria.

Furthermore, as plant flavonoids belong to phenols, our laboratory tried to explore whether similar equations could be established for phenols and found that there are also similar correlations between their lipophilicities and their inhibitory activities towards Gram-positive bacteria. However, there is no extensive applicability for phenols as their structural diversity is too great. For some specific structural types of phenols, which one is the larger or the largest compound against Gram-positive bacteria can also be roughly deduced from their lipophilicities, such as abietane diterpenoids [42,43]. In addition, the anti-MRSA activities of trimethylhydroquinone and vitamin K_3_ were successfully predicted and verified by our laboratory [44], referring to the initial assumptions of the above conclusions.

Based on the above, the antibacterial activity and mechanism of plant flavonoids against Gram-positive bacteria were diagrammatically presented in Figure 1, and some errors in Figure 9 of the published paper were incidentally corrected [8].

## 4. Materials and Methods

### 4.1. Data and Processing

Based on all the data on the plant flavonoids in the previous work [18], the data processing was reperformed. As no clear MIC value was presented for the many flavonoids used for the verification of the two regression equations in that paper [18], such as the MIC of compound **84** expressed as more than one hundred (>100 μM), these data were processed according to the following rules: (1) discard all the ambiguous data which the MICs expressed as more than a certain value; (2) for the MICs expressed as more than or equal to a certain value, the boundary value is considered as the MIC, such as the MIC for compound **69** as 636.4 μM; (3) for the MICs expressed as more than a range, the latter is considered as the MIC because it is a clear MIC value, such as the MIC for compound **73** as 888.1 μM; and (4) for the MIC expressed as a range, the average of the two boundary values is considered as the MIC because these two boundary values are the MICs of a certain flavonoid to different pathogenic strains, such as the MIC for compound **71** as 520.3 μM. Finally, based on the variation tendency of MIC, along with the lipophilicity parameters of LogP or LogD previously reported, and in view of the probable fluctuation at the determination of the MICs, the probable outliers were discarded using a scatter diagram. All the rest of the data were used as the data of this article for the subsequent analyses. The physicochemical parameter LogP or LogD_7.40_ (the log_10_ of oil/water distribution coefficient at pH 7.40) was calculated using the software ACD/Labs 6.0 (Advanced Chemistry Development, Inc., Toronto, ON, Canada).

### 4.2. Regression Analyses

For establishing the antimicrobial quantitative relationship of plant flavonoids to Gram-positive bacteria, the regression analysis between the MICs (*y*) and the physicochemical parameter LogP or LogD_7.40_ (*x*) was performed using Microsoft Excel software (Microsoft Corporation, USA), and the *r* value was also calculated. To discover more powerful evidence for supporting the antibacterial mechanism of plant flavonoids acting on the cell membrane of Gram-positive bacteria, the MIC was further transformed to the log_10_(MIC), and subsequently, the regression analysis between the log_10_(MIC) (*y*) and the physicochemical parameter LogP or LogD_7.40_ (*x*) was further achieved.

### 4.3. Statistical Analyses

To ensure that one is the most reliable, further statistical analyses were performed for all the regression equations, including those reported in the previous paper [18]. In the process, two statistical parameters, the coefficient of determination (***R*^2^**), and the residual standard deviation (***s***) were calculated, respectively, according to Equations (5) and (6).
(5)$$R^{2}=1-\frac{\sum {\left(y_{i}-\hat{y_{i}}\right)}^{2}}{\sum {\left(y_{i}-\bar{y}\right)}^{2}}$$
(6)$$s=\sqrt{\frac{\sum {\left(y_{i}-\hat{y_{i}}\right)}^{2}}{n-2}}$$
where $y_{i}$ is the MIC of a certain flavonoid *i*. Correspondingly, $\hat{y_{i}}$ is the predicted MIC of flavonoid *i*, $\bar{y}$ is the average MIC of all flavonoids in Table 1, and *n* (*n* = 92) is the number of all flavonoids.

When comparing the goodness of fit of these regression curves, the closer the *R***^2^** is to 1, the higher the goodness of fit and the closer the predicted value is to the actual one, on the whole. The smaller *s* is, the smaller the mean deviation between the predicted value and the actual one. Generally, a consistent result will be presented from these two statistical parameters.

## 5. Conclusions

In conclusion, the MICs (y) of most plant flavonoids to Gram-positive bacteria can be calculated by substituting their physicochemical parameter ACD/LogP (x) into the equation *y* = −0.1285 *x*^6^ + 0.7944 *x*^5^ + 51.785 *x*^4^ − 947.64 *x*^3^ + 6638.7 *x*^2^ − 21,273 *x* + 26,087. More reliable equations than before further proved that the lipophilicity is a key factor of plant flavonoids against Gram-positive bacteria, and more intuitive evidence powerfully confirmed the antibacterial mechanism, which is that the cell membrane is the major site of plant flavonoids acting on the Gram-positive bacteria and likely involves the damage of phospholipid bilayers. The above will greatly accelerate the discovery and application of plant flavonoids with remarkable antibacterial activity and accelerate the screening for the leading antibacterial compounds from the reported plant flavonoids. In addition, it can also provide a good reference for the structural modification and optimization of plant flavonoids if no heteroatom is introduced into their structures and can reduce trial and error. Simultaneously, all of the above provide good references for exploring the antibacterial activity and mechanisms of plant flavonoids against Gram-negative bacteria.

## Figures and Tables

**Figure 1 pharmaceuticals-15-01190-f001:**
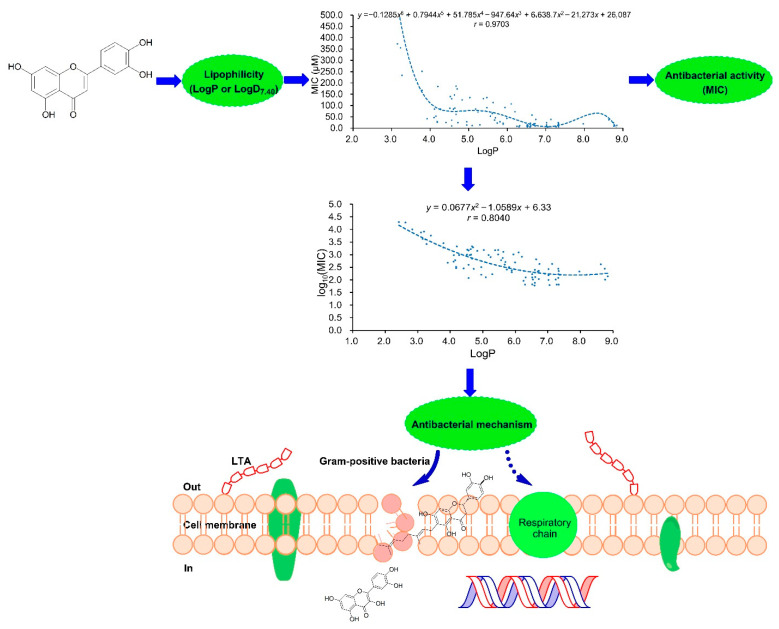
Diagrammatic presentation for the lipophilicity, the antibacterial activities, and the mechanisms of plant flavonoids. MIC, minimum inhibitory concentration; LTA, Lipoteichoic acid.

**Figure 2 pharmaceuticals-15-01190-f002:**
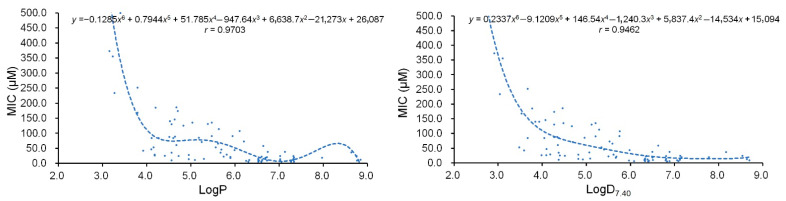
Polynomial regression analyses for the physicochemical parameter LogP or LogD_7.40_ (*x*) and the MIC (*y*) to Gram-positive bacteria, mainly including *Staphylococcus aureus*, *S. epidermidis*, or/and *Bacillus subtilis* of 92 plant flavonoids.

**Figure 3 pharmaceuticals-15-01190-f003:**
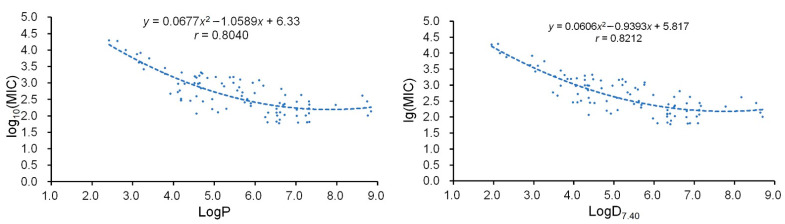
Regression analyses for the physicochemical parameter LogP or LogD_7.40_ (*x*) and the log_10_(MIC) (*y*) to Gram-positive bacteria of ninety-two plant flavonoids.

**Table 1 pharmaceuticals-15-01190-t001:** Plant flavonoids together with their structure types, physicochemical parameters, and antimicrobial activities, used for the regression analyses [18].

Compounds ^a^	Structure Types	LogP ^b^	LogD_7.40_ ^b^	MIC (μM) ^c^	Log_10_(MIC) ^c^
**2**	Dihydroflavones	5.09	4.92	11.3	1.0531
**3**	Dihydroflavones	7.02	6.8	8.85	0.9469
**4**	Dihydroflavones	5.29	5.09	14.7	1.1673
**6**	Dihydroflavones	7.02	6.81	23.7	1.3747
**7**	Dihydroflavones	4.18	4.09	25.9	1.4133
**8**	Dihydroflavones	4.18	3.98	25.9	1.4133
**9**	Dihydroflavonols	5.74	5.5	22.7	1.3560
**10**	Dihydroflavones	6.52	6.33	5.9	0.7709
**11**	Dihydroflavones	6.30	6.08	5.7	0.7559
**12**	Dihydroflavones	7.05	6.83	5.5	0.7404
**13**	Dihydroflavones	7.27	7.09	5.7	0.7559
**16**	Dihydroflavones	7.24	7.06	9.15	0.9614
**17**	Dihydroflavones	4.56	4.37	10.5	1.0212
**20**	Dihydroflavones	5.56	5.34	52.8	1.7226
**21**	Dihydroflavones	6.54	6.32	9.15	0.9614
**22**	Dihydroflavones	6.61	6.39	11.35	1.0550
**23**	Dihydroflavones	5.18	4.96	85.05	1.9297
**24**	Dihydroflavonols	6.25	5.97	8.05	0.9058
**25**	Dihydroflavones	7.02	6.81	13.65	1.1351
**26**	Dihydroflavones	7.32	7.12	10.6	1.0253
**27**	Dihydroflavones	6.72	6.51	20.4	1.3096
**28**	Dihydroflavones	3.27	3.04	233.7	2.3687
**29**	Dihydroflavones	4.60	4.38	84.4	1.9263
**30**	Dihydroflavones	4.27	4.05	84.1	1.9248
**31**	Dihydroflavones	4.67	4.46	186.4	2.2704
**32**	Dihydroflavones	6.10	5.76	107.3	2.0306
**33**	Dihydroflavones	5.63	5.29	113.6	2.0554
**34**	Flavonols	4.52	3.84	140.2	2.1467
**35**	Flavonols	4.52	3.93	140.2	2.1467
**36**	Flavonols	6.20	5.53	73	1.8633
**37**	Dihydroflavones	6.72	6.51	9.5	0.9777
**38**	Dihydroflavones	7.32	7.12	14.75	1.1688
**39**	Dihydroflavones	8.75	8.54	24.45	1.3883
**40**	Dihydroflavones	7.32	7.13	24.6	1.3909
**41**	Dihydroflavones	5.94	5.75	90.8	1.9581
**42**	Dihydroflavones	7.97	7.78	19	1.2788
**43**	Dihydroflavones	6.74	6.50	37.9	1.5786
**44**	Dihydroflavones	8.84	8.64	12.25	1.0881
**45**	Dihydroflavonols	3.79	3.67	251.75	2.4010
**46**	Dihydroflavonols	3.79	3.53	167.8	2.2248
**47**	Dihydroflavonols	3.92	3.59	42.1	1.6243
**48**	Dihydroflavonols	4.67	4.35	61	1.7853
**49**	Dihydroflavonols	4.11	3.67	84.5	1.9269
**52**	Dihydroflavonols	4.51	4.27	87.8	1.9435
**53**	Dihydroflavonols	2.42	2.11	1734.6	3.2392
**54**	Dihydroflavonols	4.64	4.34	88.3	1.9460
**55**	Dihydroflavones	6.52	6.33	11.05	1.0434
**56**	Dihydroflavones	8.76	8.70	9	0.9542
**57**	Dihydroflavones	4.72	4.51	24.25	1.3847
**58**	Dihydroflavones	6.52	6.33	14.7	1.1673
**59**	Dihydroflavones	5.89	5.67	17.75	1.2492
**60**	Dihydroflavones	5.89	5.68	21.3	1.3284
**61**	Dihydroflavones	6.60	6.35	22.05	1.3434
**62**	Dihydroflavones	5.81	5.62	28.4	1.4533
**63**	Dihydroflavones	5.81	5.62	28.4	1.4533
**64**	Dihydroflavones	4.56	4.37	35.1	1.5453
**66**	Dihydroflavones	3.19	2.96	734.6	2.8661
**67**	Flavones	4.20	3.77	184.7	2.2665
**70**	Flavonols	3.10	2.32	670.5	0.7597
**72**	Isoflavones	7.33	6.89	5.75	2.9485
**73**	Flavonols	2.83	2.16	888.1	1.5653
**75**	Dihydroflavonols	8.63	8.17	36.75	0.7284
**76**	Flavones	6.59	6.40	5.35	0.8325
**77**	Dihydroflavones	6.60	6.42	6.8	1.4518
**81**	Chalcones	4.95	4.82	28.3	1.1508
**82**	Chalcones	4.95	4.82	14.15	1.5502
**86**	Isoflavones	5.67	5.07	35.5	2.5715
**87**	Isoflavones	3.15	2.91	372.8	2.1166
**88**	Isoflavones	5.38	5.12	130.8	1.7239
**89**	Flavonols	4.15	3.48	52.95	1.0434
**91**	Dihydroisoflavane	6.32	6.32	11.05	1.4031
**92**	Dihydroisoflavane	4.41	4.4	25.3	1.4609
**93**	Dihydroisoflavane	4.18	4.18	28.9	1.7649
**94**	Other type	6.64	6.63	58.2	3.2186
**97**	Flavonols	2.62	1.95	1654.3	2.5505
**113**	Chalcones	3.23	3.10	355.2	2.6985
**114**	Chalcones	3.40	3.26	499.5	2.1318
**115**	Isoflavones	5.03	4.48	135.45	2.1649
**116**	Isoflavones	4.63	4.07	146.2	2.2399
**118**	Isoflavones	5.69	5.4	45.41	1.2940
**119**	Isoflavones	7.33	7.16	19.68	1.8510
**120**	Isoflavones	5.24	4.69	70.95	2.2398
**121**	Isoflavones	4.70	4.27	173.7	1.5786
**122**	Isoflavones	7.13	6.89	37.9	2.1149
**123**	Dihydroisoflavones	4.56	4.27	130.3	2.1318
**124**	Dihydroisoflavones	5.47	5.21	135.45	1.9557
**125**	Dihydroisoflavones	5.47	5.21	90.3	2.0964
**126**	Dihydroisoflavones	4.83	4.67	124.85	1.2765
**127**	Dihydroisoflavones	6.69	6.5	18.9	1.6375
**128**	Other type	5.99	5.98	43.4	2.5224
**130**	Other type	5.61	5.59	65.15	1.6721
**133**	Other type	4.10	4.10	47	0.7597

^a^: The chemical structures of flavonoids shown in previous work [18]. ^b^: The LogP and LogD_7.40_ values were calculated using software ACD/Labs 6.0. ^c^: MIC, minimum inhibitory concentration; here, a processed MIC of a certain flavonoid to various Gram-positives, including *Staphylococcus aureus*, *S. epidermidis*, or/and *Bacillus subtilis*, etc., was presented; log_10_(MIC) means log_10_ of MIC.

**Table 2 pharmaceuticals-15-01190-t002:** Regression equations for the correlation between the physicochemical parameter (*x*) and the antimicrobial activity (*y*) to Gram-positive bacteria of plant flavonoids ^a^.

Equation Number	Sample Numbers (*n*)	Parameters ^b^ (*x*)	Regression Equation (*r* ^c^)
(1)	92	LogP	*y =* −0.1285 *x*^6^ + 0.7944 *x*^5^ + 51.785 *x*^4^ − 947.64 *x*^3^ + 6638.7 *x*^2^ − 21,273 *x* + 26,087 (0.9703)
(2)	92	LogD_7.40_	*y* = 0.2337 *x*^6^ − 9.1209 *x*^5^ + 146.54 *x*^4^ − 1240.3 *x*^3^ + 5837.4 *x*^2^ − 14,534 *x* + 15,094 (0.9462)
(3)	66 ^d^	LogP	*y* = −1.6745 *x*^5^ + 56.143 *x*^4^ − 741.93 *x*^3^ + 4831.8 *x*^2^ − 15,531 *x* + 19,805 (0.9349)
(4)	66 ^d^	LogD_7.40_	*y* = −1.1474 *x*^5^ + 38.802 *x*^4^ − 515.39 *x*^3^ + 3361.9 *x*^2^−10,789 *x* + 13,706 (0.9309)

^a^: The antimicrobial activity (*y*) was the average MIC (or MIC_90_) of a certain flavonoid to Gram-positive bacteria, mainly including *Staphylococcus aureus*, *S. epidermidis*, and *Bacillus subtilis*. ^b^: The physicochemical parameter (*x*) was calculated using software ACD/Labs 6.0. ^c^: *r*, correlation coefficient; the significant level *α* was set as 0.01, and the critical values of *r*_0.995_ (90) and *r*_0.995_(64) were equal to 0.27 and 0.32, respectively. ^d^: The regression equations were established in previous work [18].

**Table 3 pharmaceuticals-15-01190-t003:** The goodness of fit of the regression equations ^a^.

Equation Number	Coefficient of Determination (*R*^2^)	Residual Standard Deviation (*s*)	Goodness of Fit
(1)	0.9413	68.1127	The best one
(2)	0.8949	91.1187	The better one
(3)	0.8740	89.5452	—
(4)	0.8666	92.1391	—

^a^: Equations (3) and (4) were reported in previous work [18], and therefrom, the *R***^2^** and ***s*** values were calculated from the 66 data pairs.

**Table 4 pharmaceuticals-15-01190-t004:** The relationship between the structure and the lipophilicity (LogP) and some novel structure–activity relationships of plant flavonoids.

Structural Segment	Contribution for the Lipophilicity Parameter LogP Value	The Antimicrobial Structure–Activity Relationship of Plant Flavonoids
Structural skeleton (Ring C)	(1) The LogP value for Chalcones > dihydrochalcones, flavonols > flavones, dihydroisoflavones, dihydroflavones > isoflavones, dihydroflavonols. (2) When ring C is open, the LogP values remarkably increase, such as chalcones and dihydrochalcones.	Overall consistency with that reported [7,8,13,14].
Hydroxyl group	(1) The hydroxyl group substituting on ring A rather than ring B has greater contribution for the LogP value of flavonoids.	(1) Uncertain.
	(2) Generally, the contribution of hydroxyl groups substituting on ring A for the LogP value of flavonoids as: for flavones: 7-OH > 5-OH > 5,7-di-OH; for flavonols: 5-OH ≈ 7-OH > 5,7-di-OH; for chalcone, dihydrochalcones, dihydroisoflavones, dihydroflavones, isoflavones and dihydroflavonols: 5-OH > 5,7-di-OH > 7-OH.	(2) and (3) Overall, the contributions of hydroxyl groups for antimicrobial activity were consistent with that reported [7,8,13,14], while the contributed sequence was not presented. A new SAR was proposed as follows: The hydroxyls will increase the antimicrobial activity, while the molecules must have enough lipophilicity. Otherwise, the increase in hydroxyl would reduce the antimicrobial activity. Namely, the molecular lipophilicity would likely mask the influences on antimicrobial activity from the hydroxyls.
	(3) Generally, the contribution of hydroxyl groups substituting on ring B for the LogP value of flavonoids as: 2’-OH ≥ 4’-OH (≈ 2’,4’-di-OH) > 3’,4’-di-OH (≈ 3’,4’,5’-tri-OH) > 2’,4’,5’-tri-OH > 2’,4’,6’-tri-OH.	
	(4) The LogP values will be increased a little or remain unchanged when the hydroxyl groups are methylated.	(4) Antimicrobial activity increases or not depending on the position of methylated hydroxyls and the structural subclass [7,13,14].
Isopentenyl chains	(1) The introductions of isopentenyl groups into the skeleton would remarkably increase the LogP values, while their substituted positions present no obvious influence on the LogP values. In addition, the number increase of isopentenyl units on structural skeleton will remarkably increase the LogP values. However, the dissociations of hydroxyls on structural skeleton will decrease along with the increase of isopentenyl units.	(1) Antimicrobial activity will remarkably increase, which is consistent with that reported [7,8,13,14]. However, a new SAR was proposed as follows: (1) the substituted positions of isopentenyl chains into the skeleton likely present no obvious influence on the antimicrobial activity; (2) the number increase of isopentenyl units on structural skeleton would increase the antimicrobial activity. However, too many isopentenyl units (usually, above 4) would lead to the slight decrease in antimicrobial activity. Both the above SARs were mainly summarized from the data of previous reports [18].
	(2) The introductions of the hydroxyl group into the isopentenyl side chain would sharply reduce the LogP values.	(2) Antimicrobial activity would sharply reduce, which was first summarized from the data of previous reports [18].

## Data Availability

The raw data of all compounds, which generated the data of Table 1 in this study, were reported by us in Springer Nature and are available from the link at https://www.nature.com/articles/s41598-021-90035-7 (accessed on 18 May 2021). All other data generated or analyzed during this study are included in this published article.

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
