# Peer review of "Antimicrobial Quantitative Relationship and Mechanism of Plant Flavonoids to Gram-Positive Bacteria"

_pharmaceuticals, 2022, doi:10.3390/ph15101190_

Round 1

Reviewer 1 Report

The authors are examining the antimicrobial quantitative relationship and mechanism of plant flavonoids to gram-positive bacteria. The authors do make a good argument for studying this topic since they note that there is a strong relationship between some crucial antimicrobial parameters and the biocide effect of targeted bacteria.

Overall, I think this study does a very good job showing that the goal of the study is scientific-relevant. I have only a few comments and suggestions for the Authors:

·         Abstract needs to be rewritten because it is not necessary to explain the way for obtaining statistically valued results. Please, correct it and make it better for future readers.

·         When looking at the review instructions, the guideline indicates that most references should be younger than 5 years. Looking at the list of references cited, there is a range with several that are much older than 5 years. I am not sure if there are more recent references available for some of these older references. The authors may wish to check this. Some of the older references are methodology references though.

 ·         I think the authors are missing some citations within the Introduction and would like to see these. I’d also like to see the authors be specific when a reference is used and related to only a portion of a statement and not have it at the end of the statement when the reference does not pertain to the other items in the statement

 ·         In the discussion sections, I feel that there was a low level of other research presented. There was no full context to the current research and more references in needed for comparison with the following study.

 ·         Definition of the MIC is required in the Introduction part; put adequate reference and relate to the presented study.

 ·         All Figures need to be revised; please expand the graphs and enlarge the letters

 ·         line 108 – names of microorganisms need to be italic

 ·         line 126 – names of microorganisms need to be italic

 ·         check the whole text, and uniform writing “r” value – italic or not?

 ·         line 145 – change sentence avoiding using “by us” and all other similar situations in the text (we, our...)

 ·         In the Conclusion part avoid using references and targeted some tables and figures; highlight the most important results, advantages, and disadvantages of this study, as well as future perspectives. 

Author Response

Dear Reviewer,

My co-authors and I are very grateful to you for your careful review, valuable suggestions, and great works for improving our research. We have amended the manuscript according to the issues raised by you, and have pleasure to submit the revised version, together with the response to all points, for your consideration.

Many thanks for your kind attention!

Yours sincerely,

Ganjun Yuan

Here are our responses to your comments.

Comments and Suggestions for Authors

The authors are examining the antimicrobial quantitative relationship and mechanism of plant flavonoids to gram-positive bacteria. The authors do make a good argument for studying this topic since they note that there is a strong relationship between some crucial antimicrobial parameters and the biocide effect of targeted bacteria.

Overall, I think this study does a very good job showing that the goal of the study is scientific-relevant. I have only a few comments and suggestions for the Authors:

Response: Thank you for your careful review, objective evaluations and valuable suggestions!

Point 1: Abstract needs to be rewritten because it is not necessary to explain the way for obtaining statistically valued results. Please, correct it and make it better for future readers.

Response: Thank you for your valuable suggestion!

According to your suggestion, we had deleted the way for obtaining statistically valued results. Another, we had carefully checked throughout the abstract and made some other revisions for your consideration.

Point 2: When looking at the review instructions, the guideline indicates that most references should be younger than 5 years. Looking at the list of references cited, there is a range with several that are much older than 5 years. I am not sure if there are more recent references available for some of these older references. The authors may wish to check this. Some of the older references are methodology references though.

Response: Thank you for your careful review and valuable suggestion!

According to your suggestion, we had deleted 5 older references and inserted 16 new recent ones, for more accurate citations and citing recent researches. As 12 papers (references 19-30) directly relating to the correlation analyses, which were used for our previous publication, were published before 2015, some older and directly related references still remain unchanged. This coincide with that you subconsciously reminded us that it is very important for us to respect scientists and researchers with their direct contributions, although the guideline of review instructions indicates that most references should be younger than 5 years.  

Many revisions for references and corresponding citations were performed. To avoid format and citation errors, we had completed it not all using the “Track Changes” function. All new references inserted were listed as follows for your consideration.

  1. Liang, M.; Ge, X.; Xua, H.; Ma, K.; Zhang, W.; Zan, Y.; Efferth, T.; Xue, Z.; Hua, X. Phytochemicals with activity against methicillin-resistant Staphylococcus aureus. Phytomedicine 2022, 100, 154073. https://doi.org/10.1016/j.phymed.2022.154073
  2. Farhadi, F.; Khameneh, B.; Iranshahi, M.; Iranshahy, M. Antibacterial activity of flavonoids and their structure-activity relationship: An update review. Phytother. Res. 2019, 33, 13-40. https://doi.org/10.1002/ptr.6208
  3. Tan, Z.; Deng, J.; Ye, Q.; Zhang, Z. The antibacterial activity of natural-derived flavonoids. Curr. Top. Med. Chem. 2022, 22, 1009-1019. https://doi.org/10.2174/1568026622666220221110506
  4. Song, L.; Hu, X.; Ren, X.; Liu, J.; Liu, X. Antibacterial modes of herbal flavonoids combat resistant bacteria. Front. Pharmacol. 2022, 13, 873374. https://doi.org/10.3389/fphar.2022.873374
  5. Wu, S.C.; Yang, Z.Q.; Liu, F.; Peng, W.J.; Qu, S.Q.; Li, Q.; Song, X.B.; Zhu, K.; Shen, J.Z. Antibacterial effect and mode of action of flavonoids from licorice against methicillin-resistant Staphylococcus aureus. Front. Microbiol. 2019, 10, 2489. https://doi.org/10.3389/fmicb.2019.02489
  6. Zhou, K.; Yang, S.; Li, S.M. Naturally occurring prenylated chalcones from plants: structural diversity, distribution, activities and biosynthesis. Nat. Prod. Rep. 2021, 38, 2236-2260. https://doi.org/10.1039/d0np00083c
  7. Shamsudin, N.F.; Ahmed, Q.U.; Mahmood, S.; Ali Shah, S.A.; Khatib, A.; Mukhtar, S.; Alsharif, M.A.; Parveen, H.; Zakaria, Z.A. Antibacterial effects of flavonoids and their structure-activity relationship study: a comparative interpretation. Molecules 2022, 27, 1149. https://doi.org/10.3390/molecules27041149
  8. Echeverría, J.; Opazo, J.; Mendoza, L.; Urzúa, A.; Wilkens, M. Structure-activity and lipophilicity relationships of selected antibacterial natural flavones and flavanones of Chilean flora. Molecules 2017, 22, 608. https://doi.org/10.3390/molecules22040608
  9. Magozwi, D.K.; Dinala, M.; Mokwana, N.; Siwe-Noundou, X.; Krause, R.W.M.; Sonopo, M.; McGaw, L.J.; Augustyn, W.A.; Tembu, V.J. Flavonoids from the Genus Euphorbia: isolation, structure, pharmacological activities and structure-activity relationships. Pharmaceuticals 2021, 14, 428. https://doi.org/10.3390/ph14050428
  10. Araya-Cloutier, C.; Vincken, J.P.; van de Schans, M.G.; Hageman, J.; Schaftenaar, G.; den Besten, H.M.; Gruppen, H. QSAR-based molecular signatures of prenylated (iso) flavonoids underlying antimicrobial potency against and membrane-disruption in Gram positive and Gram negative bacteria. Sci. Rep. 2018, 8, 1–14.
  11. Chen, X.; Mukwaya, E.; Wong, M.S.; Zhang, Y. A systematic review on biological activities of prenylated flavonoids. Pharm. Biol. 2014, 52, 655-660. https://doi.org/10.3109/13880209.2013.853809
  12. 34. Qiu, F.; Meng, L.; Chen, J.; Jin, H.; Jiang, L. In vitro activity of five flavones from Scutellaria baicalensisin combination with Cefazolin against methicillin resistant Staphylococcus aureus (MRSA). Chem. Res. 2016, 25, 2214–2219.
  13. Donadio, G.; Mensitieri, F.; Santoro, V.; Parisi, V.; Bellone, M.L.; de Tommasi, N.; Izzo, V.; Dal Piaz, F. Interactions with microbial proteins driving the antibacterial activity of flavonoids. Pharmaceutics 2021, 13, 660. https://doi.org/10.3390/pharmaceutics13050660
  14. Rammohan, A.; Bhaskar, B.V.; Venkateswarlu, N.; Rao, V.L.; Gunasekar, D.; Zyryanov, G.V. Isolation of flavonoids from the flowers of Rhynchosia beddomei Baker as prominent antimicrobial agents and molecular docking. Microb. Pathog. 2019, 136, 103667. https://doi.org/10.1016/j.micpath.2019.103667.
  15. Mohamed, M.S.; Abdelkader, K.; Gomaa, H.A.M.; Batubara, A.S.; Gamal, M.; Sayed, A.M. Mechanistic study of the antibacterial potential of the prenylated flavonoid auriculasin against Escherichia coli. Arch. Pharm. (Weinheim) 2022, e2200360. https://doi.org/10.1002/ardp.202200360
  16. Fang, Y.; Lu, Y.; Zang, X.; Wu, T.; Qi, X.; Pan, S.; Xu, X. 3D-QSAR and docking studies of flavonoids as potent Escherichia coli inhibitors. Sci. Rep. 2016, 6, 23634. https://doi.org/10.1038/srep23634

Point 3: I think the authors are missing some citations within the Introduction and would like to see these. I’d also like to see the authors be when a reference is used and related to only a portion of a statement and not have it at the end of the statement when the reference does not pertain to the other items in the statement.

Response: Thank you for your kind reminder and valuable suggestion!

According to your suggestion, we had revised section Introduction, and inserted some new references. Another, we had adjusted some citations for more specific and clearer expression logic. The new inserted references can be found in the response to point 2 (references 5, 7,9-12, and 14-17), for your consideration.

Point 4: In the discussion sections, I feel that there was a low level of other research presented. There was no full context to the current research and more references in needed for comparison with the following study.

Response: Thank you for your kind reminder and valuable suggestion!

According to your suggestion, we had revised section Discussion, inserted some new references and supplemented related context in section Discussion of revised manuscript, for comparison with the following study. The new inserted references can be found in the response to point 2 (references 17, 31, 34, 36, 38, 40 and 41) for your consideration. Moreover, we had properly cited these references and developed appropriate discussions around the new references.

Point 5: Definition of the MIC is required in the Introduction part; put adequate reference and relate to the presented study.

Response: Thank you for your kind reminder and valuable suggestion!

The minimum inhibitory concentration (MIC) is an indicator of antibacterial effect, which was similar to that the LogP or LogD is an indicator of lipophilicity. According to your suggestion, we had presented a proper explanation as “minimum inhibitory concentration (MIC, an indicator of antibacterial effect)”, and which indicated the relation to the presented study and what the MIC is. Since everyone for professionals in this field knows, we had not presented a definition and related references.  

Point 6: All Figures need to be revised; please expand the graphs and enlarge the letters

Response: Thank you for your kind reminder and valuable suggestion!

According to your suggestion, we had already used larger font for coordinates and regression equations in Figure 1. Simultaneously, the graphs of Figures 2 and 3 were expanded, and correspondingly the letters were enlarged.

Point 7: line 108 – names of microorganisms need to be italic

Response: Thank you for your kind reminder and valuable suggestion!

According to your suggestion, we had already revised them.

Point 8:  line 126 – names of microorganisms need to be italic

Response: Thank you for your kind reminder and valuable suggestion!

According to your suggestion, we had already revised them.

Point 9: check the whole text, and uniform writing “r” value – italic or not?

Response: Thank you for your kind reminder and careful review!

According to your suggestion, we had already checked all the font of “ r ”, and kept all of them in italic.

Point 10: line 145 – change sentence avoiding using “by us” and all other similar situations in the text (we, our...)

Response: Thank you for your kind reminder and valuable suggestion!

According to your suggestion, we had changed the sentences to avoid using “by us” and “we”, and reduced the use of the word “our” as possible as we can. To avoid ambiguity and keep more accurate description of the facts, four expressions (including two for our laboratory) for “our” still remained, and they can be found at Lines 26, 50, 297 and 423 in the revised manuscript.

Point 11:  In the Conclusion part avoid using references and targeted some tables and figures; highlight the most important results, advantages, and disadvantages of this study, as well as future perspectives. 

Response: Thank you for your kind reminder and valuable suggestion!

According to your suggestion, we had already deleted the reference and the targeted figure, and moved them to section Discussion. Simultaneously, some other revision had been performed in section Conclusion.

Some other revisions:

Besides above revisions, we had carefully performed other extensive revisions throughout the manuscript including references, linguistic edit, expression, spelling, and grammar, etc. for your consideration.

Thank you very much for your review and great help to improve our works!

Reviewer 2 Report

 Dear author(s):

Antimicrobial quantitative relationship and mechanism of plant flavonoids to gram-positive bacteria.

Antimicrobial resistance (AMR) poses a serious threat to human health, and new antimicrobial agents are need. Plant flavonoids have been increasingly paid attention to, for their antibacterial activities, enhancing the antibacterial activity of antimicrobials, and reversing the AMR. Moreover, the lipophilicity is a key factor of plant flavonoids against bacteria.

After an exhaustive revision, the manuscript (pharmaceuticals-1907481) is Accept after minor revision.

In general, the study is closely connected to the journal's objectives. 

The study is very interesting.

The abstract is good, since it is no so long, it is no close to introduction.

The introduction is concise and precise, and it has updated references until 2022.

The authors have written an introductions, with concise and precise information.

The section discussion is good, includes a description of the results, the lines correspond to results section and on explication of results, and on comparison with others studies, and on explication (discussion) of the results obtained with respect to other studies.

In the following pages, I give a detailed revision of the manuscript.

The section "3. Discussion", some sentence need to improve "Probably, this equation can be also used for the crude calculation of the MIC values when the logP values are more than 8.9.".

“(1) The logP values of hydroxyl groups on ring A is larger than those on ring B.”

“(2) Ring A: the logP values”

“(3) Ring B: generally, the logP values 2'-OH≥4'-OH≈2',4'-OH>3',4'-OH (3',4',5'-OH)>2',4',5'-OH>2',4',6'-OH”

20. Hatano, T.; Shintani, Y.; Aga, Y.; Shiota, S.; Tsuchiya, T.; Yoshida, T. Phenolic constituents of Licorice. VIII.1) Structures of glicophenone and glicoisoflavanone, and effects of licorice phenolics on methicillin-resistant Staphylococcus aureus. Chem. Pharm. Bull. 2000, 48, 1286–1292; doi:10.1248/cpb.48.1286.

21. Veitch, N.C.; Grayer, R.J. Flavonoids and their glycosides, including anthocyanins. Nat. Prod. Rep. 2004, 21, 539–573.

Author Response

Dear Reviewer,

My co-authors and I are very grateful to you again for your careful review, valuable suggestions, and great works for improving our research. We have amended the manuscript according to your suggestions, and have pleasure to submit the revised version, together with the response to all points, for your consideration.

Many thanks for your kind attention!

Yours sincerely,

Ganjun Yuan

Here are our answers to your comments.

Point 1: After an exhaustive revision, the manuscript (pharmaceuticals-1907481) is Accept after minor revision.

In general, the study is closely connected to the journal's objectives.

The study is very interesting.

The abstract is good, since it is no so long, it is no close to introduction.

The introduction is concise and precise, and it has updated references until 2022.

The authors have written an introduction, with concise and precise information.

The section discussion is good, includes a description of the results, the lines correspond to results section and on explication of results, and on comparison with others studies, and on explication (discussion) of the results obtained with respect to other studies.

Response: Thanks for your careful review, very nice evaluation, and great help to improve our work!

Point 2: In the following pages, I give a detailed revision of the manuscript.

The section "3. Discussion", some sentence need to improve "Probably, this equation can be also used for the crude calculation of the MIC values when the logP values are more than 8.9.".

Response: Thank you for your kind reminder and valuable suggestion!

According to your suggestion, we had already revised them as “Probably, when the logP values of plant flavonoids are more than 8.9, this equation can be still used for the crude calculation for their MIC values against gram-positive pathogenic bacteria”, for your consideration.

(1) The logP values of hydroxyl groups on ring A is larger than those on ring B.

Response: Thank you for your kind reminder!

We had already revised them as “The hydroxyl group substituting on ring A than ring B has greater contribution for the logP value of flavonoids”, for your consideration.

(2) Ring A: the logP values”

Response: Thank you for your kind reminder!

We had already revised them as “Generally, the contribution of hydroxyl groups substituting on ring A for the logP value of flavonoids as:”, for your consideration.

(3) Ring B: generally, the logP values 2'-OH≥4'-OH≈2',4'-OH>3',4'-OH (3',4',5'-OH)>2',4',5'-OH>2',4',6'-OH

Response: Thank you for your kind reminder!

We had already revised them as “Generally, the contribution of hydroxyl groups substituting on ring B for the logP value of flavonoids as: 2'-OH ≥ 4'-OH (≈ 2',4'-di-OH) > 3',4'-di-OH (≈ 3',4',5'-tri-OH) > 2',4',5'-tri-OH > 2',4',6'-tri-OH.”, for your consideration.

(4) 20. Hatano, T.; Shintani, Y.; Aga, Y.; Shiota, S.; Tsuchiya, T.; Yoshida, T. Phenolic constituents of Licorice. VIII.1) Structures of glicophenone and glicoisoflavanone, and effects of licorice phenolics on methicillin-resistant Staphylococcus aureus. Chem. Pharm. Bull. 2000, 48, 1286–1292; doi:10.1248/cpb.48.1286.

Response: Thank you for your kind reminder!

We had already checked it, and revised it as “30. Hatano, T.; Shintani, Y.; Aga, Y.; Shiota, S.; Tsuchiya, T.; Yoshida, T. Phenolic constituents of licorice. VIII. Structures of glicophenone and glicoisoflavanone, and effects of licorice phenolics on methicillin-resistant Staphylococcus aureus. Chem. Pharm. Bull. 2000, 48, 1286–1292. https://doi.org/10.1248/cpb.48.1286”, for your consideration.

(5) 21. Veitch, N.C.; Grayer, R.J. Flavonoids and their glycosides, including anthocyanins. Nat. Prod. Rep. 2004, 21, 539–573.

Response: Thank you for your kind reminder!

We had already checked it, and revised it as “32. Veitch, N.C.; Grayer, R.J. Flavonoids and their glycosides, including anthocyanins. Nat. Prod. Rep., 2011, 28, 1626–1695.”, for your consideration.

Some other revisions:

Besides above revisions, we had carefully performed other extensive revisions throughout the manuscript including references, linguistic edit, expression, spelling, and grammar, etc. for your consideration.

Thank you very much for your review and great help to improve our works!

Round 2

Reviewer 1 Report

/